# Cytoplasmic Expression of TP53INP2 Modulated by Demethylase FTO and Mutant NPM1 Promotes Autophagy in Leukemia Cells

**DOI:** 10.3390/ijms24021624

**Published:** 2023-01-13

**Authors:** Junpeng Huang, Minghui Sun, Yonghong Tao, Jun Ren, Meixi Peng, Yipei Jing, Qiaoling Xiao, Jing Yang, Can Lin, Li Lei, Zailin Yang, Ling Zhang

**Affiliations:** 1Key Laboratory of Laboratory Medical Diagnostics Designated by the Ministry of Education, College of Laboratory Medicine, Chongqing Medical University, No. 1, Yixueyuan Road, Chongqing 400016, China; 2Department of Hematology-Oncology, Chongqing Key Laboratory of Translational Research for Cancer Metastasis and Individualized Treatment, Cancer Hospital, Chongqing University, Chongqing 400030, China

**Keywords:** acute myeloid leukemia, nucleophosmin 1, TP53INP2, autophagy, N^6^-methyladenosine

## Abstract

Acute myeloid leukemia (AML) with a *nucleophosmin 1* (*NPM1*) mutation is a unique subtype of adult leukemia. Recent studies show that *NPM1*-mutated AML has high autophagy activity. However, the mechanism for upholding the high autophagic level is still not fully elucidated. In this study, we first identified that tumor protein p53 inducible nuclear protein 2 (TP53INP2) was highly expressed and cytoplasmically localized in *NPM1*-mutated AML cells. Subsequent data showed that the expression of TP53INP2 was upregulated by fat mass and obesity-associated protein (FTO)-mediated m^6^A modification. Meanwhile, TP53INP2 was delocalized to the cytoplasm by interacting with NPM1 mutants. Functionally, cytoplasmic TP53INP2 enhanced autophagy activity by promoting the interaction of microtubule-associated protein 1 light chain 3 (LC3) - autophagy-related 7 (ATG7) and further facilitated the survival of leukemia cells. Taken together, our study indicates that TP53INP2 plays an oncogenic role in maintaining the high autophagy activity of *NPM1*-mutated AML and provides further insight into autophagy-targeted therapy of this leukemia subtype.

## 1. Introduction

Acute myeloid leukemia (AML) is a hematological malignancy carrying accumulated genetic abnormalities [1]. *Nucleophosmin 1* (*NPM1*) mutation is one of the most common genetic alterations in AML, detected in approximately 30–35% of adult AML [2]. Notably, *NPM1* mutations endow NPM1 mutants with stronger nuclear export ability that results in the cytoplasmic delocalization of NPM1 mutants, which is thought to serve a critical role in leukemogenesis [3]. Accumulating studies support *NPM1* mutation as a driver lesion of myeloproliferation and hematological cancer [4,5]. However, a study in an *NPM1*-mutated transgenic mice model indicated that *NPM1* mutation alone is insufficient to initiate leukemia [6], suggesting the need for exploring other underlying oncogenic events in *NPM1*-mutated AML. Of note, emerging studies indicate that dysregulated autophagy is a non-negligible driving factor leading to leukemogenesis and malignant progression [7].

Autophagy is an intracellular catabolic degradation process that supports metabolic adaptation and nutrient recycling [8]. Plenty of studies demonstrate that dysfunctional autophagy contributes to tumorigenesis including leukemia [9,10]. Increasing evidence indicates that targeting autophagy holds promise for enhancing the therapeutic benefit of AML treatment [11]. In particular, our group has previously reported that *NPM1*-mutated AML cells have high autophagy activity [12]. Partial mechanistic studies reveal that pyruvate kinase isoenzyme M2 (PKM2) [13] and TNF receptor-associated factor 6 (TRAF6) [14] participate in the autophagic activation of leukemia cells. Recently, several novel insights related to autophagy regulation have been proposed [15,16]. Methyltransferase METTL3-mediated N^6^-methyladenosine (m^6^A) modification inhibits cell autophagy activity by reducing the PTEN stability and further results in the chemoresistance of leukemia cells [17]. Despite these existing studies, the underlying mechanism for autophagic activation has not been fully elucidated in *NPM1*-mutated leukemia.

Tumor protein p53 inducible nuclear protein 2 (TP53INP2) is a newly discovered scaffold protein that plays an important role in the process of autophagy activation [18]. *TP53INP2* is first identified on human chromosome band 20q11.2 by in situ hybridization [19]. It is homologous to TP53INP1 which acts as a cofactor of p53 to regulate apoptosis and cell cycle [20,21]. In fact, TP53INP2 fails to participate in the regulation of p53 like TP53INP1 [22]. As a nuclear protein, TP53INP2 mainly participates in the transcriptional regulation of nuclear hormone receptors to sustain muscle or adipose wasting [23]. Recently, Mauvezin et al. reported that nuclear TP53INP2 delocalizes to the cytoplasm under nutrient starvation, stimulating autophagosome formation and accelerating the degradation of stable proteins in mammalian and drosophila cells [24]. Up to now, accumulating studies have shown that TP53INP2-mediated autophagy participates in the occurrence of many diseases, including adiposity [25], diabetes [26], and tumors [27]. However, the role of TP53INP2 in leukemia, especially in *NPM1*-mutated leukemia, has been unknown.

In our study, we first determined the high expression and cytoplasmic localization of TP53INP2 in *NPM1*-mutated leukemic cells. Mechanistically, TP53INP2 was upregulated by demethylase FTO-mediated m^6^A modification and delocalized into the cytoplasm by interacting with NPM1-mA. More importantly, cytoplasmic TP53INP2 could promote the interaction of LC3-ATG7 to enhance autophagy activity and further facilitate leukemia cell proliferation. Collectively, our findings uncover the oncogenic role of TP53INP2 in facilitating the high autophagy activity of *NPM1*-mutated AML and propose further insights into the autophagy-targeted therapy of this distinct leukemic entity.

## 2. Results

### 2.1. Identification of the Autophagy-Related Genes in NPM1-Mutated AML

To identify the dysregulated autophagy-related genes in *NPM1*-mutated leukemia, we first investigated the differentially expressed genes (DEGs) between *NPM1*-mutated AML cases and *NPM1*-unmutated AML cases from three public datasets (TCGA, GSE15434, and Beat-AML) and took the intersection of DEGs from three datasets. A total of 110 candidate DEGs were obtained (Figure 1A). Then, these candidate DEGs were used to intersect with the autophagy-related genes derived from the HADb database and three autophagy-related genes (*TP53INP2*, *NKX2-3*, and *TUSC1*) were screened out (Figure 1B). Next, the heatmap showed that *TP53INP2* and *NKX2-3* were significantly gathered in *NPM1*-mutated AML cases whereas *TUSC1* was mainly expressed in *NPM1*-unmutated AML cases (Figure 1C). Additionally, upregulated *TP53INP2* and *NKX2-3* were identified in *NPM1*-mutated AML cases of GSE15434 datasets (Figure 1D). Furthermore, gene set enrichment analysis (GSEA) showed that the positive regulation of the autophagy and autophagosome assembly process were significantly enriched in the high *TP53INP2* expression group (Figure 1E,F). However, autophagy-associated pathways were not obtained in the enrichment results of *NKX2-3* and *TUSC1*. Finally, the Kaplan–Meier analysis indicated that AML patients with high *TP53INP2* expression had poorer overall survival, while the other two genes showed no prognostic value (Figure 1G). In addition, we investigated the expression profile of *TP53INP1* which is highly homologous to *TP53INP2*, whereas the result showed no significant difference between *NPM1*-mutated and *NPM1*-unmutated AML patients. Thus, we focused on *TP53INP2* for further investigations in this study.

### 2.2. TP53INP2 Is Aberrantly Expressed in NPM1-Mutated Leukemia Cells

To further determine the *TP53INP2* expression in *NPM1*-mutated leukemia, we acquired the RNA-seq data of *TP53INP2* in a group of leukemia-lymphoma cell lines in the CCLE database. The results showed that there is higher *TP53INP2* expression in OCI-AML3 cells with NPM1 mutation type A(NPM1-mA) compared to other leukemia cell lines without NPM1-mA (Figure 2A). Then, the expression of *TP53INP2* transcripts in leukemia blasts derived from AML patients was detected. The results showed that *TP53INP2* was dramatically upregulated in *NPM1*-mutated leukemia patients (*n* = 21) compared with *NPM1*-unmutated leukemia patients (*n* = 18) (Figure 2B). Furthermore, qRT-PCR and Western blot assays indicated that *TP53INP2* mRNA and proteins were highly expressed in NPM1-mA positive OCI-AML3 cells (Figure 2C,D). Besides, the cellular localization of TP53INP2 was detected by immunofluorescence staining. The results revealed that TP53INP2 was largely expressed in the cytoplasm of OCI-AML3 cells and mostly expressed in the nucleus of OCI-AML2 and THP-1 cells (Figure 2E). Similarly, cytoplasmic localization of TP53INP2 in OCI-AML3 cells was also confirmed by nucleus and cytoplasm fraction isolation assays (Figure 2F–H). These observations demonstrate the high expression and cytoplasmic localization of TP53INP2 in *NPM1*-mutated leukemia cells.

### 2.3. TP53INP2 Is Upregulated by FTO-Mediated m^6^A Modification

Next, we explored the molecular mechanism underlying the high expression of *TP53INP2* in *NPM1*-mutated leukemia cells. Considering NPM1-mA is a driver of AML [28], we first knocked down NPM1-mA to investigate whether it is involved in the regulation of *TP53INP2*. The results showed that NPM1 knockdown decreased *TP53INP2* mRNA and protein levels in OCI-AML3 cells (Figure 3A,B). Inversely, NPM1-mA overexpression increased *TP53INP2* mRNA and protein levels in OCI-AML2 cells, but NPM1 wild-type (NPM1-wt) overexpression resulted in no significant changes (Figure 3C,D). Nevertheless, based on the fact that NPM1 mutants cannot directly regulate gene transcription [29], we deduced that other molecular events participate in the regulation of *TP53INP2* transcripts. Considering our recent studies showed that NPM1-mA can stabilize m^6^A demethylase FTO to participate in the transcriptional regulation of target genes [30], we investigated whether FTO-mediated demethylation regulates the expression of *TP53INP2*. Initially, several m^6^A methylated sites in the *TP53INP2* mRNA were identified using the sequence-based RNA adenosine methylation site predictor (SRAMP) (Figure 3E). Moreover, FTO was predicted to bind in the CDS region of the *TP53INP2* sequence using the Starbase database (Figure 3F). Subsequently, the interaction between *TP53INP2* mRNA and FTO was verified by RNA-binding protein immunoprecipitation (RIP) assays (Figure 3G). Next, silencing FTO significantly reduced *TP53INP2* mRNA levels (Figure 3H). Similarly, meclofenamic acid (MA) and FB23-2, two selective inhibitors for FTO activity, were applied to treat OCI-AML3 cells, and also observed that *TP53INP2* mRNA levels were decreased (Figure 3I). Furthermore, both FTO knockdown and activity inhibition increased the m^6^A modification levels in the *TP53INP2* mRNA (Figure 3J,K) and resulted in the decreased stability of *TP53INP2* mRNA (Figure 3L). These results indicate that the expression of *TP53INP2* is upregulated by FTO-mediated m^6^A modification in *NPM1*-mutated leukemia cells.

### 2.4. TP53INP2 Is Delocalized into the Cytoplasm by Interacting with NPM1-mA

Followingly, we looked into the reason that TP53INP2 is largely located in the cytoplasm of *NPM1*-mutated leukemia cells. Considering that NPM1-mA engages in the translocation of various nuclear proteins [31], we first investigated whether the delocalization of TP53INP2 is related to NPM1-mA. The results showed that NPM1-mA overexpression increased cytoplasmic TP53INP2 and decreased nuclear TP53INP2 in OCI-AML2 and THP-1 cells (Figure 4A,B). Meanwhile, immunofluorescence analysis showed that both NPM1-mA and TP53INP2 co-localized in the cytoplasm of OCI-AML3 cells (Figure 4C). Furthermore, the interaction of endogenous TP53INP2 and NPM1-mA was detected by Co-Immunoprecipitation (Co-IP) analysis (Figure 4D,E). Additionally, the combination between TP53INP2 and NPM1-mA was also confirmed in HEK293T cells co-transfected with HA-TP53INP2 and Flag-NPM1-mA plasmids (Figure 4F,G). Finally, cytoplasmic TP53INP2 was relocalized to the nucleus following treatment with KPT-330, an inhibitor of nuclear exportation of NPM1-mA (Figure 4H). Similar results of TP53INP2 relocalization were also shown in immunofluorescence analysis (Figure 4I). The findings demonstrate that TP53INP2 is delocalized into the cytoplasm of leukemia cells by interacting with NPM1-mA.

### 2.5. TP53INP2 Facilitates Autophagy Activity by Promoting the Interaction of LC3-ATG7

To clarify the role of TP53INP2 in the autophagy process of *NPM1*-mutated leukemia cells, we first silenced TP53INP2 to observe the changes in autophagic level. The results showed that TP53INP2 depletion decreased LC3-II/I levels and increased p62 levels (Figure 5A). Similar results were obtained by immunofluorescence analysis, as indicated by the accumulated LC3 puncta in OCI-AML3 cells (Figure 5B). Conversely, TP53INP2 overexpression elevated LC3II/I ratio and diminished p62 levels in OCI-AML2 cells (Figure 5C). In addition, an increased number of LC3 puncta was observed by immunofluorescence analysis (Figure 5D). Given that autophagy-related gene 7 (ATG7) interacts with LC3 and is an important link in autophagosome formation [32], we explored the role of TP53INP2 on the interaction of LC3-ATG7. The String analysis indicated the potential interaction of TP53INP2, LC3, and ATG7 (Figure 5E). Subsequently, the results of a Co-IP assay confirmed that TP53INP2 interacted with LC3 and ATG7 in OCI-AML3 cells (Figure 5F–H). Furthermore, immunoprecipitation analysis revealed that the silencing of TP53INP2 decreased the amount of ATG7 proteins pulled down by LC3 (Figure 5I). In contrast, overexpressing TP53INP2 increased the amount of ATG7 proteins pulled down by LC3 (Figure 5J). In addition, immunofluorescent results showed that the depletion of TP53INP2 diminished the accumulation of LC3 and ATG7 puncta (Figure 5K). In conclusion, these data suggest that TP53INP2 enhances autophagy by promoting the interaction of LC3-ATG7.

### 2.6. TP53INP2-Mediated Autophagy Is Vital for Leukemia Cell Survival

Finally, we explored whether TP53INP2-enhanced autophagy is required for leukemia cell survival. The autophagy activator rapamycin and autophagy inhibitor 3-methyladenine (3-MA) was used to treat leukemia cells, respectively. The results showed that rapamycin treatment reversed the decreased LC3-II/I levels and the increased p62 levels in TP53INP2-silenced OCI-AML3 cells (Figure 6A). Furthermore, TP53INP2 knockdown inhibited the cell proliferation, and then the effect was partially rescued by rapamycin treatment (Figure 6B,C). In contrast, 3-MA treatment disrupted the enhanced autophagy activity and cell proliferation ability in TP53INP2-enforced OCI-AML2 cells (Figure 6D–F). In the end, we performed a rescue experiment to further probe the effect of TP53INP2 on NPM1-mA-maintained autophagy and cell survival. The data indicated that NPM1-mA knockdown reduced LC3II/I ratio and increased p62 protein levels, and then these changes were successfully rescued by TP53INP2 overexpression (Figure 6G). Moreover, CCK-8 assays showed that overexpressing TP53INP2 partially retrieved the effect of NPM1-mA depletion on cell proliferation (Figure 6H). Collectively, these results indicate that TP53INP2-enhanced autophagy is essential for the survival of NPM1-positive leukemia cells.

## 3. Discussion

Accumulating evidence has supported that dysfunctional autophagy is closely associated with leukemogenesis and development [10]. However, the mechanism of high autophagic levels in *NPM1*-mutated leukemia has not yet been fully clarified. Here, we used bioinformatics analysis to screen out the autophagy-associated dysregulated gene *TP53INP2* in *NPM1*-mutated leukemia. Subsequent data revealed that TP53INP2 was upregulated by demethylase FTO-mediated m^6^A modification and delocalized to the cytoplasm by interacting with NPM1-mA. Importantly, cytoplasmic TP53INP2 enhanced autophagy activity by promoting the interaction of LC3-ATG7, thereby maintaining leukemia cell survival (Figure 7).

In the current study, we first utilized bioinformatics analysis to ascertain the dysregulated genes associated with autophagy in *NPM1*-mutated leukemia and obtained three genes including *TUSC1*, *NKX2-3*, and *TP53INP2*. Given that survival analysis showed the high expression of *TP53INP2* was related to poor prognosis, we mainly focused on *TP53INP2* in this study. Then, we confirmed the high expression of *TP53INP2* in OCI-AML3 cells and primary blasts. Indeed, RNA-seq data from another study also indicated the upregulation of *TP53INP2* in *NPM1*-mutated leukemia cells [33]. Followingly, we explored the mechanism of high *TP53INP2* expression. Initially, we revealed that knockdown and overexpression of NPM1-mA affected the level of *TP53INP2* transcripts. Considering NPM1-mA cannot directly regulate the transcription of target genes [29] but NPM1-mA can stabilize demethylase FTO to participate in transcriptional regulation [30], we investigated whether FTO-mediated demethylation modulates *TP53INP2* expression. Then, the binding of *TP53INP2* mRNA and FTO was confirmed by RIP experiments. Moreover, m^6^A RIP and RNA stability assays further identified that FTO upregulated *TP53INP2* expression by reducing the m^6^A modification in the *TP53INP2* transcripts. In line with our finding, a recent study indicated that the inhibition of FTO activity reduces *PPARG* mRNA levels in bone marrow mesenchymal stem cells [34]. In fact, up to 40% of all mRNA expression is subject to regulation by FTO-mediated demethylation in AML [35]. It has been well documented that m^6^A modification must be identified by reader proteins, which leads to different destinies of target RNA [36,37]. Qing et al. reported that reader protein YTHDF2 recognizes m^6^A modification in *PFKP* and *LDH*, and further upregulates their post-transcriptional expression in AML [38]. Thus, further studies are required to determine which reader protein identifies the m^6^A modification in *TP53INP2*. In this work, we also observed that cytoplasmic TP53INP2 was more abundant in leukemia cells with NPM1-mA than those without NPM1-mA. Furthermore, the results of Co-IP assays confirmed that NPM1 mutants interacted with TP53INP2 and gave rise to the cytoplasmic delocalization of TP53INP2. Indeed, plenty of studies indicated that NPM1 mutants act as a chaperone protein to induce cytoplasmic delocalization of many nuclear proteins including ARF [39], CTCF [40], HEXIM1 [41], PU.1 [42]. Of note, the chaperone activity of NPM1 mutants relies on its hydrophobic N-terminal region [31]. Therefore, the exact domain necessitated for the binding between NPM1 mutants and TP53INP2 demands to be further investigated.

Next, we attempted to explore the biological function of TP53INP2 in *NPM1*-mutated leukemia cells. First, we observed that TP53INP2 knockdown reduced the LC3 II/I ratio and increased p62 levels while TP53INP2 overexpression performed the opposite effect, which indicated that TP53INP2 contributed to the autophagy activation. Our idea was supported by Yang et al. and reported that the downregulation of TP53INP2 induced by oxidative stress suppresses the autophagy degradation pathway in bone mesenchymal stem cells [43]. Additionally, it is said that TP53INP2 overexpression induces the activation of colon cancer cell autophagy via the HEDGEHOG signaling pathway [27]. Next, we explored the molecular mechanism by which TP53INP2 activated autophagy in leukemia. The results demonstrated that TP53INP2 could directly interact with ATG7 and further stabilize the binding of ATG7 and LC3, contributing to autophagosome formation. Parallelly, Liu et al. reported that TP53INP2 acts as a scaffold protein to promote the interaction of ATG7 and LC3 in HEK293T cells [44]. In addition, it is quoted that TP53INP2 chaperones deacetylated nuclear LC3 to the cytoplasm to promote macroautophagy [45]. Recently, TP53INP2 has been identified as a novel autophagic adaptor that recruits ubiquitin and ubiquitinated proteins to autophagosomes for degradation [46]. Nowak et al. proposed that TP53INP2 could interact with the transmembrane protein VMP1 to promote the formation of autophagosomes [47]. It is ordinarily recognized that autophagy is a multistep process in which distinct sets of proteins control different steps [48]. Therefore, further studies are needed to investigate whether TP53INP2 regulates other autophagy-related components to engage in the autophagy process in *NPM1*-mutated leukemia.

In the end, we assessed the role of TP53INP2-mediated autophagy in maintaining the survival of leukemia cells. The data uncovered that the autophagy inhibitor 3-MA treatment could abrogate the enhancement of TP53INP2-mediated autophagic activation on leukemic cell survival. Furthermore, NPM1-mA knockdown impaired autophagy activity and cell proliferation, and TP53INP2 overexpression partially reversed the effect of NPM1-mA deficiency. These observations confirmed that TP53INP2-modulated autophagy was essential for the survival of *NPM1*-mutated leukemia cells. In keeping with our findings, TP53INP2 is reported to induce autophagy activation to promote pancreatic cancer progression [49]. Additionally, Hu et al. claimed that TP53INP2-mediated basal autophagy makes liposarcoma cells more resistant to bortezomib-induced growth suppression [50]. Apart from being involved in autophagy, TP53INP2 can modulate epithelial-to-mesenchymal transition via the GSK-3β/β-Catenin/Snail1 pathway in bladder cancer cells [51]. Moreover, a recent study has uncovered a novel role for TP53INP2, which acts as an accelerator of caspase-8 activation, favoring death receptor-mediated apoptosis [52]. Therefore, a deeper exploration of other biological functions of TP53INP2 contributes to revealing its distinct role in *NPM1*-mutated leukemia. Of course, further work is necessitated to clarify the role of TP53INP2-mediated autophagy in mouse knock-in models that mimic human *NPM1*-mutated leukemia. In a word, our study first reveals that the high expression of TP53INP2 mediated by FTO and cytoplasmic delocalization of TP53INP2 modulated by NPM1-mA jointly facilitates autophagic pro-survival, which provides a new theoretical basis for *NPM1*-mutated AML therapy.

## 4. Materials and Methods

### 4.1. Data Analysis of TCGA, GEO, Beat-AML, CCLE, and HADb Databases

The RNA-seq data and clinical characteristics of leukemia patients were obtained from The Cancer Genome Atlas (TCGA, *n* = 170, http://www.cancergenome.nih.gov, accessed on 12 October 2021), the Beat-AML database (*n* = 448, http://vizome.org/additional_figures_BeatAML.html, accessed on 12 October 2021), and the NCBI Gene Expression Omnibus (GEO) under the accession number GSE15434 (*n* = 251, https://www.ncbi.nlm.nih.gov/gds, accessed on 12 October 2021). The expression data of target genes in leukemia cell lines were downloaded from the Cancer Cell Line Encyclopedia (CCLE, https://depmap.org/portal/, accessed on 15 November 2021). The autophagy-associated genes were obtained through the Human Autophagy Database (HADb, http:// www.autophagy.lu/, accessed on 18 October 2021). We utilized R software (Version 3.6.3) to analyze the above data. First, the limma package was used to acquire differentially expressed genes (DEGs) (*p* value < 0.05 as a cut-off criterion). Then, these DEGs were visualized by R packages. The Veen diagrams were obtained using the R package VennDiagram. The heatmap was created using pheatmap and ggplot2 packages. For Gene Set Enrichment Analysis (GSEA), R package clusterProfiler was used to visualize the autophagy-associated signal pathways. At last, the Kaplan–Meier survival analyses were performed by R package survival.

### 4.2. Cell Lines and Culture

The OCI-AML3 and OCI-AML2, human myeloid leukemia cell lines, were bought from Deutsche Sammlung von Mikroorganismen und Zellkulturen GmbH (DSMZ, Braunschweig, NI, Germany) and cultivated in RPMI-1640 medium (Thermo Fisher Scientific, Waltham, MA, USA, #11875093) containing 10% fetal bovine serum (FBS) (Thermo Fisher Scientific, #10099141C) and 1% penicillin-streptomycin solution (P-S) (Beyotime, Shanghai, China, #C0222). The THP-1, NB4, and KG-1a, human myeloid leukemia cell lines, were acquired from the American Type Culture Collection (ATCC, Manassas, VA, USA) and grew in RPMI-1640 with 10% FBS and 1% P-S. The human embryonic kidney (HEK) 293T cells were obtained from the ATCC and cultivated in DMEM with 10% FBS (Thermo Fisher Scientific, #10091155) and 1% P-S. The aforementioned cells were incubated at 37 °C with 5% CO_2_.

### 4.3. Clinical Samples

The newly diagnosed AML patient samples derived from bone marrow and peripheral blood were provided by the Chongqing University Cancer Hospital. The study was approved by the Ethics Committee of Chongqing Medical University, according to the Declaration of Helsinki. Mononuclear cells were acquired by the use of Ficoll Lymphocyte Separation Solution (Hao Yang Biological Manufacture Co, Tianjin, China, #TBD2013CHU05), referring to the manufacturer’s guidance. The clinical characteristics of leukemia patients were shown in Table 1.

### 4.4. Lentiviral Vectors and Cell Infection

The lentivirus-mediated short hairpin RNA (shRNA) was applied to the knockdown analysis. The targeted sequences were as bellow: sh*NPM1*#1: 5′-GCCGACAAAGAT TATCACTTT-3′, sh*NPM1*#2: 5′-AGCAAGGTTCCACAGAAAA-3′; sh*TP53INP2*#1: 5′-CCGGTCCAAGAACCAGAGCAG-3′, sh*TP53INP2*#2: 5′-CGCCTTCGTGTCG GAGGAGGA-3′; sh*FTO*: 5′-GACAAAGCCTAACCTACTT-3′. The shRNA vectors targeting *NPM1* were bought from GenePharma (Shanghai, China), and shRNA vectors targeting TP53INP2 were purchased from GeneChem (Shanghai, China). The lentiviral vectors expressing NPM1 wild-type (NPM1-wt) and NPM1 mutation type A (NPM1-mA) were provided by GeneChem and applied to overexpression analysis. Leukemia cells were seeded in a 24-well plate and then infected with the lentivirus and additional HitransG P (GeneChem, #REVG005) for 48 h. The stably infected cell lines were obtained after puromycin selection for 7 days.

### 4.5. Plasmid Constructs and Cell Transfection

The plasmids coding Flag-NPM1-mA were bestowed by Dr. CJ Sherr (Genetics and Tumor Cell Biology, St, Jude Children’s Research Hospital, Memphis, TN, USA). The pcDNA3.1 vectors expressing TP53INP2 were obtained from Genecreate (Wuhan, China). The leukemic cells were seeded onto a 24-well plate and transfected with the constructed plasmids via the use of Lipofectamine 2000^TM^ Transfection Reagent (Invitrogen, Carlsbad, CA, USA, #11668500) for 48 h.

### 4.6. Quantitative Real-Time PCR (qRT-PCR)

The total RNAs were extracted by the use of the TRIzol reagent (Takara, Kyoto, Japan) and transcribed into cDNA using PrimeScript^TM^ RT Master Mix (Takara, #RR036A). A quantitative real-time PCR (qRT-PCR) was performed using TB Green^TM^ Premix Ex Taq^TM^ II (Tli RNaseH Plus) (Takara, #RR820A) on a CFX Connect^TM^ real-time system (Bio-Rad, Hercules, CA, USA), according to the manufacturer’s protocols. Cycling conditions were the 30 s at 95 °C for the preliminary denaturation, and amplification was operated with 39 cycles of 5 s at 95 °C, 30 s at 58 °C, 20 s at 72 °C, and lastly 10 min at 72 °C for the extension. The mRNA levels were evaluated through the 2^−ΔΔCt^ method and normalized by β-actin. Details about the primer sequences were shown in Table 2.

### 4.7. Western Blot Analysis

To inspect the target protein expression, Western blot analysis was carried out. The collected cells were cleaned three times and cracked in cell lysis buffer (Beyotime, #P0013C) with protease inhibitors (Bimake, Houston, TX, USA, #B14001). Insoluble material was removed through centrifugation at 13,300 rpm at 4 °C for 30 min, and the protein concentrations were measured using the BCA Protein Assay Kit (Beyotime, #P0010S). Subsequently, total proteins from each sample were isolated on a 12% sodium dodecyl sulfate-polyacrylamide gel electrophoresis (SDS-PAGE) and then shifted to polyvinylidene fluoride (PVDF) membranes (Bio-Rad, #1620177). The membranes were blocked in 5% non-fat milk. The PVDF membranes were incubated with the primary antibodies overnight at 4 °C. HRP-conjugated goat anti-rabbit IgG (Biosharp, Beijing, China, #BL003A) and HRP-conjugated goat anti-mouse IgG (Biosharp, #BL001A) were used as secondary antibodies at 1:5000 dilution. Detection was visualized by using an enhanced chemiluminescence solution. Other antibodies applied in this study were as follows: Anti-p62 Rabbit Recombinant mAb (#A5180, 1:1000) were acquired from Bimake; Anti-LC3B Rabbit pAb (#381544, 1:1000), Flag-tag-HRP Mouse mAb (#700002, 1:1000) were purchased from ZEN-BIOSCIENCE (Chengdu, China); Anti-TP53INP2 Rabbit pAb (#LS-C743322, 1:500) was purchased from Lifespan Biosciences (Seattle, WA, USA); Anti-NPM1 mutant Rabbit pAb (#PA1-46356,1:1000) was purchased from Thermo Fisher Scientific; Anti-Histone H3 Rabbit pAb (#ab1791,1:1000) and anti-NPM1 Rabbit mAb (#ab52644, 1:1000) were purchased from Abcam (Cambridge, United Kingdom); Anti-β-actin Mouse mAb (#TA-09, 1:1000) and anti-GAPDH (#TA-08, 1:1000) Mouse mAb were obtained from ZSGB-BIO (Beijing, China). Normal Rabbit IgG (#2729S, 1:1000), anti-HA-Tag Mouse mAb (#2367, 1:1000), and anti-ATG7 Rabbit mAb (#8558S, 1:1000) were bought from Cell Signaling Technology (Danvers, MA, USA).

### 4.8. Immunofluorescence

Cell fluorescence was detected with a double-label multiplex immunofluorescence kit (AiFang Biological, Hunan, China, #AFIHC023). The cells were stained with TYR-520 or TYR-570 fluorescent dye for 15 min and washed three times with PBS according to the manufacturer’s instructions. 4′,6-diamino-2-phenylindole (DAPI, Beyotime, China) was utilized for the nucleus staining. Cells were detected utilizing a fluorescence microscope (Nikon, Tokyo, Japan).

### 4.9. Reagents Treatment

The inhibitors and activators used in this paper were as follows: meclofenamic acid (MA) (Selleck, Houston, TX, USA, #S4295) and FB23-2 (Selleck, #S8837), two FTO demethylase activity inhibitors; Selinexor (KPT-330) (Selleck, #S7252), a selective CRM-1/XPO-1 inhibitor; Rapamycin (Selleck, #S1039), an autophagy activator; 3-Methyladenine (3-MA) (Selleck, #S2767), an autophagy inhibitor.

### 4.10. RNA-Binding Protein Immunoprecipitation (RIP) Assays

The EZMagna RIP Kit (Millipore, Burlington, MA, USA) was applied to investigate the interaction between TP53INP2 mRNA and FTO. The cells were rinsed twice with PBS and lysed in RIP lysis buffer supplemented with protease and RNase inhibitors. Then the extraction was incubated overnight at 4 °C with magnetic beads that were coated with an anti-FTO Rabbit pAb (Cell Signaling Technology, 3687S), an anti-m^6^A Rabbit pAb (Abcam, #286164) or a control IgG (Cell Signaling Technology, #2729S). Next, the protein A/G beads were cleaned, and the complex was mixed with Proteinase K to remove proteins. Finally, the target RNA was separated from the immunoprecipitated assay, and qRT-PCR was conducted.

### 4.11. mRNA Stability Assays

To examine the mRNA stability in leukemia cells, 5 μg/mL of Actinomycin D (MedChemExpress, Monmouth Junction, NJ, USA, #HY-17559) was added into the cell culture medium to disturb RNA transcription. The cells were collected at selected time points, and RNA was extracted for qRT-PCR.

### 4.12. Nucleus and Cytoplasm Fraction Isolation Assays

The nuclear and cytoplasmic fractions of the leukemic cells were separated by the use of the Nuclear and Cytoplasmic Protein Extraction Kit (Beyotime, #P0028). The harvested and washed cells were lysed in the Cytoplasmic Protein Extraction Reagent A of the Kit. After being vortexed and rested on ice for 20 min, the Cytoplasmic Protein Extraction Reagent B was added. The cytoplasmic protein was obtained by centrifugation at 15,000× *g* at 4 °C for 10 min. After the supernatant was isolated, the sediment was lysed with Nuclear Protein Extraction Reagent on ice for 30 min. The nuclear protein was obtained by centrifugation at 15,000× *g* at 4 °C for 10 min. Finally, the extracted product was detected by western blot assay, respectively.

### 4.13. Co-Immunoprecipitation Assays

The cells were lysed using the IP lysis buffer (Beyotime, #P0013) with protease inhibitors after being harvested and washed. The total cell lysate was obtained after centrifugation at 14,000× *g* at 4 °C for 30 min. Next, the A/G beads were washed twice with PBST and then incubated with a negative control IgG and special antibodies including anti-TP53INP2 antibody (Lifespan Biosciences, #LS-C743322), anti-NPM1 mutant antibody (Thermo Fisher Scientific, #PA1-46356), anti-LC3B antibody (ZEN-BIOSCIENCE, #350140) and anti-ATG7 antibody (Bimake, #A5121) for 1 h. After being washed with PBST three times again, the protein A/G beads coated with antibodies were incubated with total cell lysate on a rotating shaker at 4 °C all night. Protein A/G beads bound to immune complexes were washed twice with PBST and then boiled in 2 × SDS-PAGE Loading Buffer (Beyotime, #P0015B) for five minutes. The product was obtained and immunoblotted after the beads were removed by magnetic adsorption.

### 4.14. Cell Counting Kit-8 (CCK-8) Assays

The proliferation of cells was detected by CCK-8 assay. The lentivirus-infected cells or plasmid-transfected cells were plated into 96-well plates at 4 × 10^3^ cells per well and then react with corresponding reagents. Next, cells were cultured in RPMI-1640 medium including 10% FBS. At selected time points (0–72 h) after seeding, 10 µL CCK8 solution (Solarbio, Beijing, China, #CA1210) was put into each well, and the plate was put into a constant temperature water box at 37 °C for 2-3 h in darkness. The absorbance value was detected with a microplate reader (BioTeck, Palo Alto, CA, USA).

### 4.15. 5-Ethynyl-2′-Deoxyuridine (EdU) Assays

The cell proliferation was analyzed using a 5-Ethynyl-2′-Deoxyuridine (EdU) assay using the EdU Cell Proliferation Kit with Alexa Fluor 594 (Beyotime, #C0078S) to detect cell viability. First, the cells were seeded to a 6-well plate at 1 × 10^6^ cells per well and then 10 μM EdU was added in each well at 37 °C for 3 h. After being rinsed twice, the cells were prepared as a suspension and spread on a slide, and then fixed in 4% paraformaldehyde for 15 min Next, the slides were covered with a click reaction solution for 30 min in the dark. Subsequently, the cells were treated using 0.5% Triton X-100 for 15 min, and DAPI was applied to label the cell nuclei for 30 min. Cell proliferation was evaluated by using the ratio of EdU-stained cells (with red fluorescence) to DAPI-stained cells (with blue fluorescence). Images were analyzed using a fluorescence microscope (Nikon, Tokyo, Japan) at 100× magnification.

### 4.16. Statistical Analysis

The results were verified by three independent experiments at least. The data were normally distributed and expressed as the mean ± standard deviation (SD). The statistical analysis was conducted with GraphPad software. (Version 8.0). The statistical significance was evaluated using unpaired Student’s *t*-tests (two groups) and the one-way analysis of variance (more than two groups). The *p*-value < 0.05 was assessed as statistically significant (* *p* < 0.05, ** *p* < 0.01, *** *p* < 0.001). No significance was defined as ns.

## Figures and Tables

**Figure 1 ijms-24-01624-f001:**
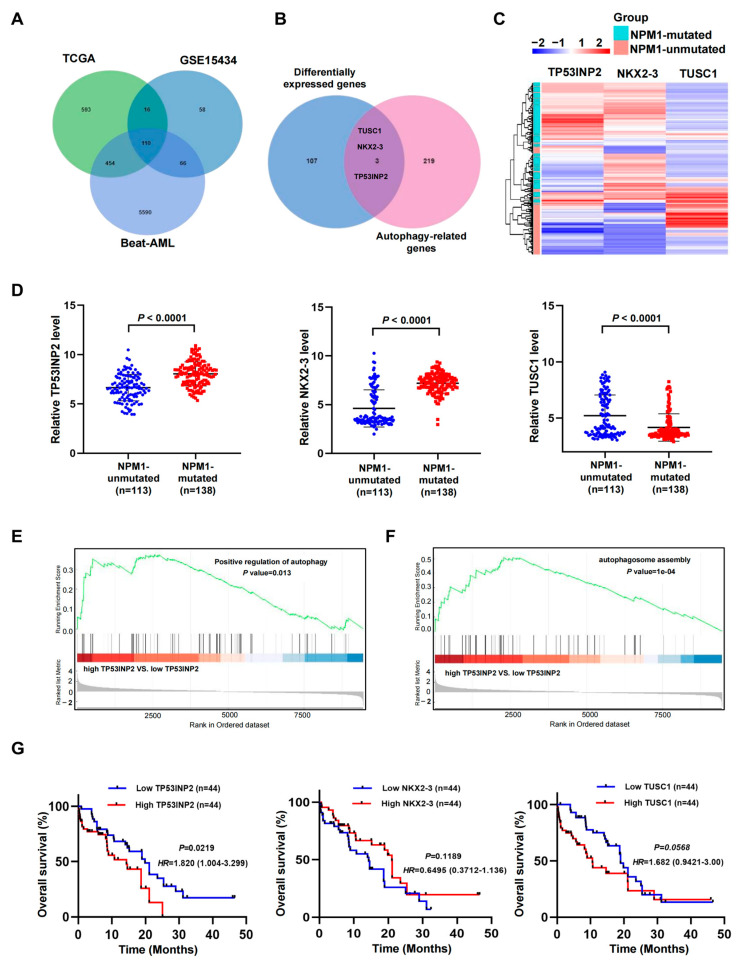
Identification of the autophagy-related genes in *NPM1*-mutated AML. (**A**) Venn diagram showing the overlap among the differentially expressed genes (*NPM1*-mutated AML vs *NPM1*-unmutated AML) obtained from three public datasets (TCGA, GSE15434, and Beat-AML). (**B**) Venn diagram showing the overlap among the differentially expressed genes and autophagy-related genes. (**C**) Heatmaps of *TP53INP2*, *NKX2-3*, and *TUSC1* genes in AML patients with *NPM1* mutations and without *NPM1* mutations from GSE15434 (*n* = 251) datasets. (**D**) The expression patterns of *TP53INP2*, *NKX2-3*, and *TUSC1* transcripts in *NPM1*-mutated AML cases compared to *NPM1*-unmutated AML cases from GSE15434 datasets. (**E**,**F**) GSEA plots showing differential enrichment of genes related to autophagosome reassembly and positive regulation of autophagy. The data were from Beat-AML. (**G**) Survival analysis based on *TP53INP2*, *NKX2-3*, and *TUSU1* mRNA expression level in primary *NPM1*-mutated AML samples of Beat-AML. Survival analysis was conducted by the use of Kaplan–Meier method.

**Figure 2 ijms-24-01624-f002:**
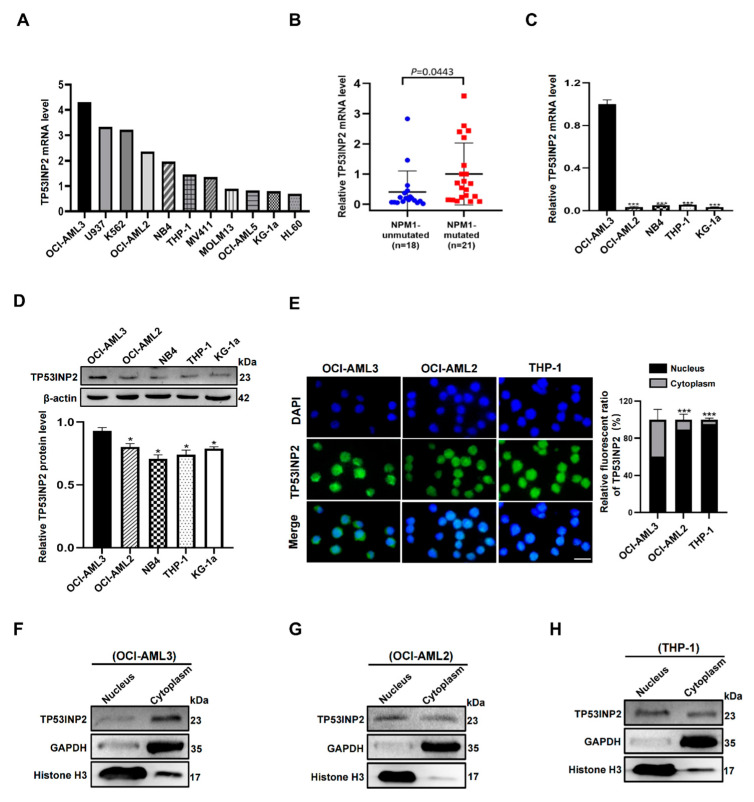
High expression and cytoplasmic localization of TP53INP2 in *NPM1*-mutated leukemia cells. (**A**) The expression of *TP53INP2* in leukemia-lymphoma cell lines from the CCLE database. (**B**,**C**) qRT-PCR analysis of relative *TP53INP2* mRNA expression in primary AML blasts and human AML cell lines. (**D**) Western blot analysis of TP53INP2 protein levels in human AML cell lines. β-actin was used as the control. The bar graph showed relative protein levels. (**E**) immunofluorescence analysis (×400) of TP53INP2 localization in OCI-AML3 cells versus OCI-AML2 and THP-1 cells. Cytoplasmic versus nuclear TP53INP2 in OCI-AML3, OCI-AML2, and THP-1 cells were quantified by Image J software. Scale bar: 50 μm. (**F**–**H**) Western blot analysis of cytoplasmic and nuclear TP53INP2 protein levels in OCI-AML3, OCI-AML2, and THP-1 cells. GAPDH was used as the cytoplasmic control. Histone H3 was used as the nuclear control. Data are presented as the mean ± SD of three independent results. * *p* < 0.05, *** *p* < 0.001.

**Figure 3 ijms-24-01624-f003:**
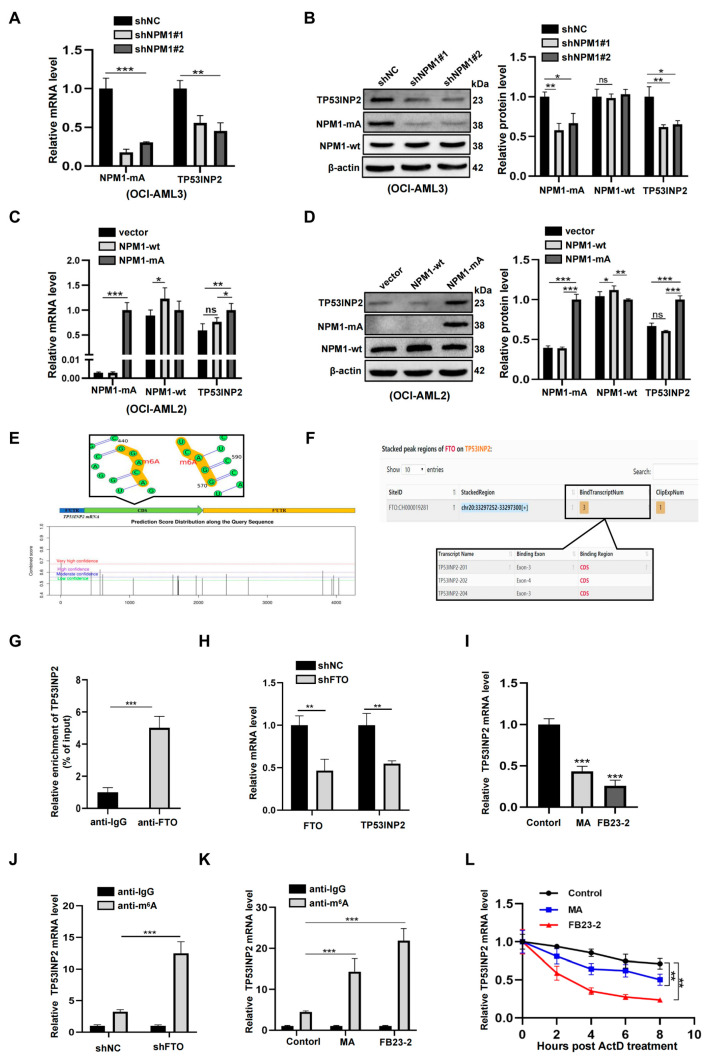
FTO-mediated m^6^A modification upregulates TP53INP2 expression. (**A**,**C**) qRT-PCR analysis of *TP53INP2*, *NPM1-wt*, and *NPM1*-*mA* mRNA levels in NPM1-mA-silenced OCI-AML3 and NPM1-mA-enforced OCI-AML2 cells. (**B**,**D**) Western blot analysis of TP53INP2, NPM1-wt, and NPM1-mA protein levels in NPM1-mA-silenced OCI-AML3 and NPM1-mA-enforced OCI-AML2 cells. β-actin was used as the control. The bar graph presented relative protein levels. (**E**,**F**) Prediction score of m^6^A distribution in *TP53INP2* mRNA sequence using SRAMP and prediction of the binding sites between *TP53INP2* mRNA and demethylase FTO using Starbase. (**G**) RIP assay was used to determine the binding between *TP53INP2* mRNA and FTO. (**H**,**I**) qRT-PCR analysis of *TP53INP2* mRNA levels after silencing FTO and inhibiting FTO activity using 20 μM FB23-2 and 50 μM MA for 24 h. (**J**,**K**) m^6^A RIP assay was used to determine the m^6^A levels in *TP53INP2* mRNA. (**L**) qRT-PCR following the addition of ActD was used to detect *TP53INP2* mRNA stability. Data are presented as the mean ± SD of three independent results. * *p* < 0.05, ** *p* < 0.01, *** *p* < 0.001. ns. indicates no significant difference.

**Figure 4 ijms-24-01624-f004:**
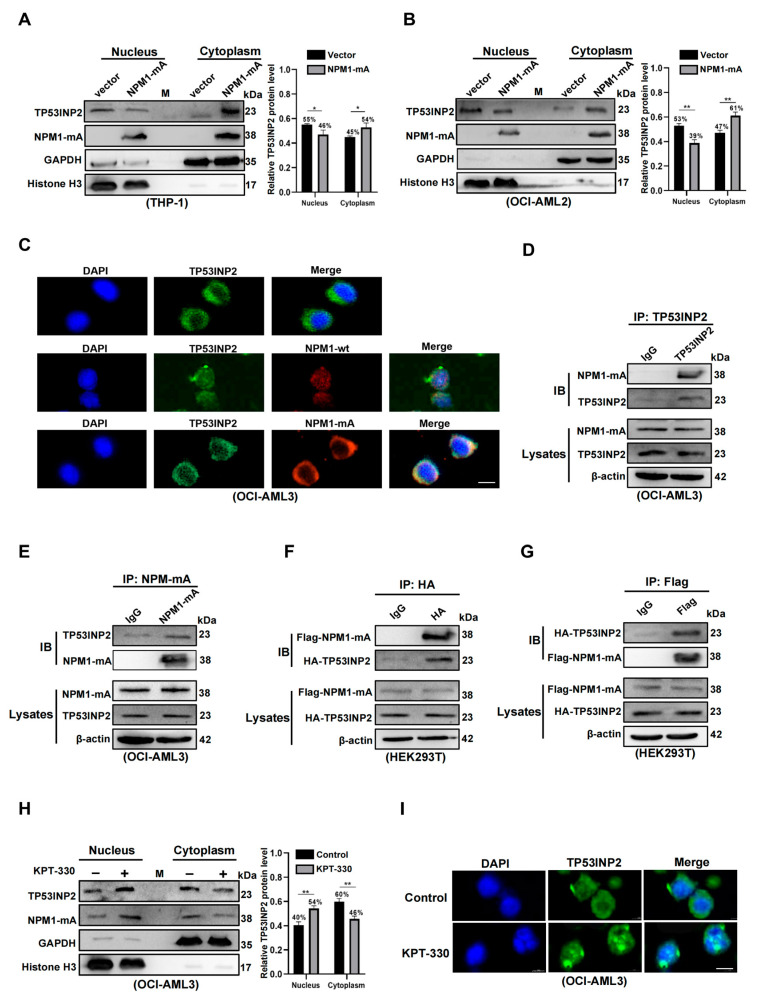
NPM1-mA interacts with TP53INP2 and leads to the cytoplasmic delocalization of TP53INP2. (**A**,**B**) Western blot analysis of cytoplasmic and nuclear TP53INP2, cytoplasmic and nuclear NPM1-mA protein levels in NPM1-mA-enforced OCI-AML2 and THP-1 cells. The bar graph exhibited the relative protein levels. (**C**) Immunofluorescence analysis (×600) of TP53INP2, NPM1-mA, and NPM1-wt localization in OCI-AML3 cells. Scale bar: 25 μm. (**D**,**E**) Total protein extracts from OCI-AML3 cells were immunoprecipitated with anti-NPM1-mA antibodies or anti-TP53INP2 antibodies, followed by immunoblotting for TP53INP2 and NPM1-mA, respectively. (**F**,**G**) The HEK293T cells were co-transfected with HA-TP53INP2 and Flag-NPM1-mA plasmids, and then cell lysates were immunoprecipitated with anti-HA antibodies or anti-Flag antibodies, followed by immunoblotting for HA-TP53INP2 and Flag-NPM1-mA. (**H**) Western blot analysis of cytoplasmic and nuclear TP53INP2, cytoplasmic and nuclear NPM1-mA protein levels in OCI-AML3 cells treated with 2 μM KPT-330 for 10 h. KPT-330 is a selective inhibitor of NPM1-mA export. (**I**) Immunofluorescence analysis (×600) of TP53INP2 localization in OCI-AML3 cells treated with 2 μM KPT-330 for 10 h. Scale bar: 25 μm. Data are presented as the mean ± SD of three independent results. * *p* < 0.05, ** *p* < 0.01.

**Figure 5 ijms-24-01624-f005:**
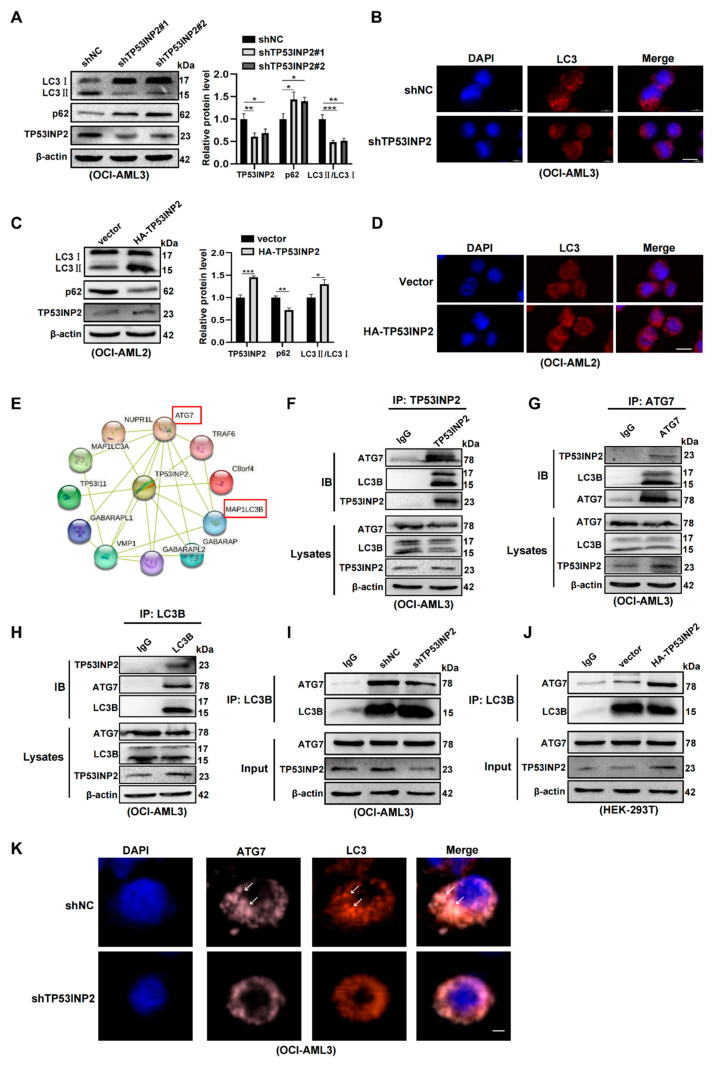
TP53INP2 promotes the interaction of LC3-ATG7 to enhance autophagy. (**A**,**C**) Western blot analysis of LC3 II/I and p62 expression in TP53INP2-silenced OCI-AML3 cells and TP53INP2-enforced OCI-AML2 cells. β-actin was used as the internal control. The bar graph displayed relative protein levels. (**B**,**D**) Immunofluorescence analysis of LC3 puncta in TP53INP2-silenced OCI-AML3 cells and TP53INP2-enforced OCI-AML2 cells. Scale bar: 25 μm. (**E**) String analysis of protein interaction network for TP53INP2. (**F**–**H**) Protein extracts from OCI-AML3 cells were immunoprecipitated with anti-TP53INP2 antibodies, anti-LC3B antibodies, or anti-ATG7 antibodies, followed by immunoblotting for LC3B, ATG7, and TP53INP2, respectively. (**I**,**J**) Western blot analysis of ATG7 levels immunoprecipitated by equal amounts of LC3B in TP53INP2-silenced OCI-AML3 cells and TP53INP2-enforced HEK-293T cells. (**K**) Immunofluorescence analysis (×600) of LC3-ATG7 puncta in TP53INP2-silenced OCI-AML3 cells. The white arrow represents the puncta of LC3-ATG7. Scale bar: 25 μm. The data are presented as the mean ± SD of three independent experiments. * *p* < 0.05, ** *p* < 0.01, *** *p* < 0.001.

**Figure 6 ijms-24-01624-f006:**
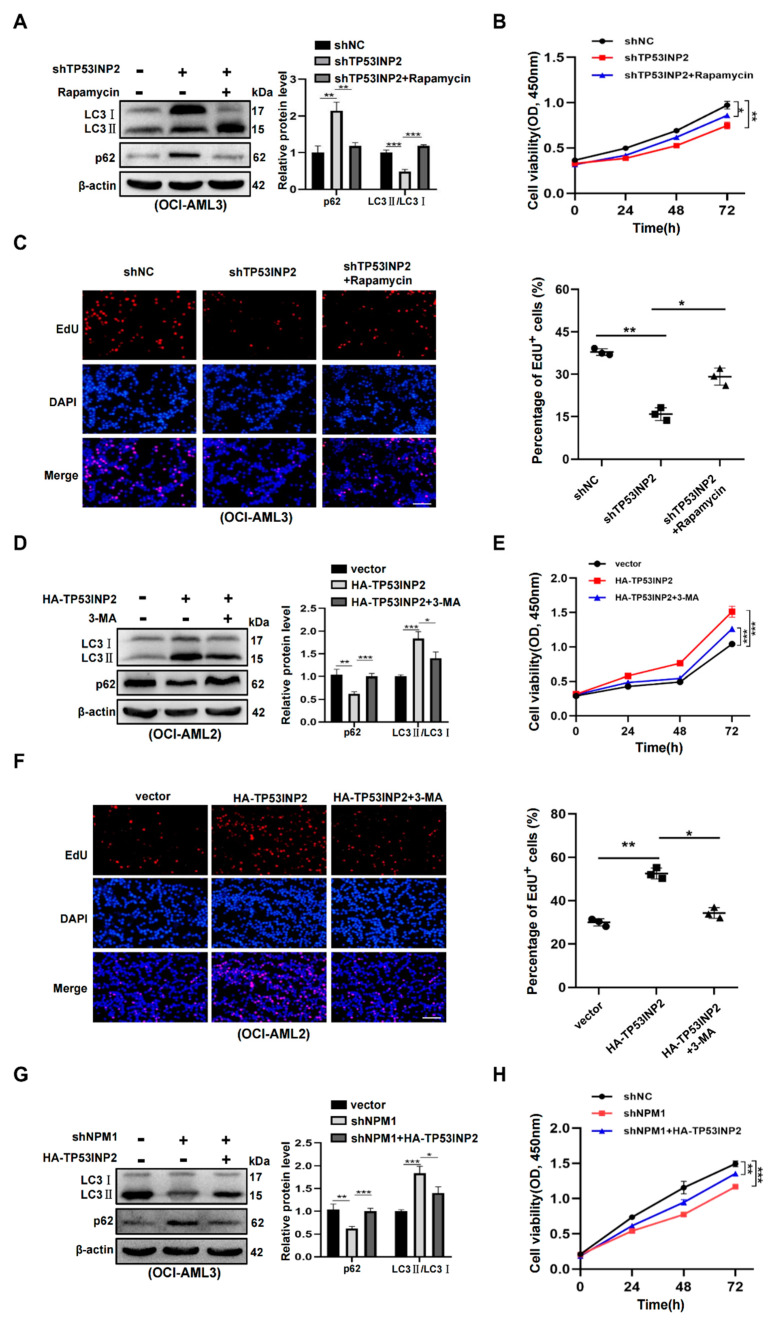
TP53INP2-mediated autophagy plays an essential role in leukemia cell survival. (**A**,**D**) Western blot analysis of LC3 II/I and p62 protein expression in TP53INP2-silenced OCI-AML3 cells treated with 5 µM rapamycin for 10 h, and TP53INP2-enforced OCI-AML2 cells treated with 2 mM 3-MA for 10 h, respectively. The bar graph presented relative protein levels. (**B**,**E**) CCK-8 assay of cell growth in TP53INP2-silenced OCI-AML3 cells treated with 5 µM rapamycin for 10 h and TP53INP2-enforced OCI-AML2 cells treated with 2 mM 3-MA for 10 h, respectively. (**C**,**F**) EdU analysis of cell proliferation in TP53INP2-silenced OCI-AML3 cells and TP53INP2-enforced OCI-AML2 cells. The bar graph showed the percentage of EdU-positive cells. Scale bar: 100 μm. (**G**) Western blot analysis of LC3II/I and p62 expression in the NPM1-mA-silenced OCI-AML3 cells following TP53INP2 overexpression. (**H**) CCK-8 analysis of cell growth ability in NPM1-mA-silenced OCI-AML3 cells following TP53INP2 overexpression. The data are presented as the mean ± SD of three independent experiments. * *p* < 0.05, ** *p* < 0.01, *** *p* < 0.001.

**Figure 7 ijms-24-01624-f007:**
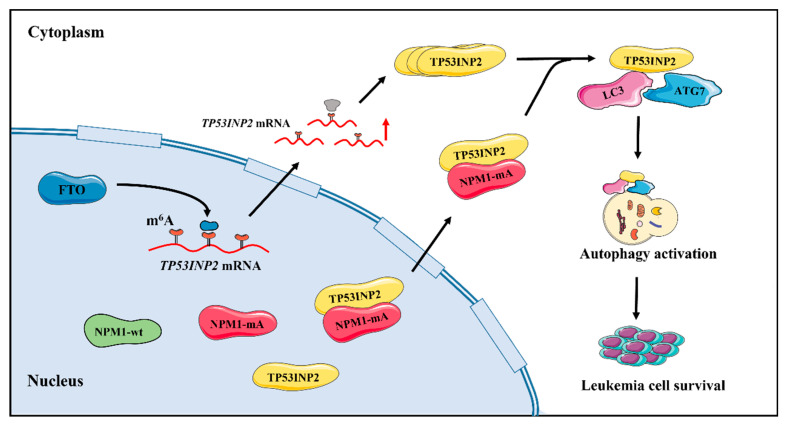
Schematic diagram describing the functional significance of cytoplasmic TP53INP2-mediated autophagy in *NPM1*-mutated leukemia cells. TP53INP2 is upregulated by FTO-mediated m^6^A modification and delocalized into the cytoplasm by NPM1-mA. Cytoplasmic TP53INP2 enhances autophagy activity by facilitating the interaction of LC3-ATG7 and further promotes the survival of leukemia cells.

**Table 1 ijms-24-01624-t001:** Clinical characteristics of newly diagnosed AML patients.

Characteristics	Median (Range)	No. of Cases
**Gender**		
Female		20
Male		19
Total		39
**Median age (years)**	51.0 (16.0–82.0)	
Younger than 40 y		12
40–60 y		19
Older than 60 y		8
**Median WBC counts (×10^9^)**	51.1 (0.7–260.8)	
**Median platelets (×10^9^)**	40.0 (3.0–176.0)	
**AML FAB subtype**		
M1		2
M2		7
M3		7
M4		6
M5		11
Other subtypes		6
**Karyotype**		
Normal		21
t (8,21)		3
t (15,17)		7
inv (16)		1
Unknown		7
**Gene mutations**		
*FLT3-ITD*		12
** *NPM1* **		21
*IDH1/IDH2*		8
*DNMT3A*		9
*WT1*		19

AML, acute myeloid leukemia; y, year old; WBC, white blood cell; FAB (French-American-British) classification, a classification of acute leukemia produced by three-nation-joint collaboration.

**Table 2 ijms-24-01624-t002:** The PCR primer sequences for each gene used in this study.

Genes	Sequences (5′-3′)
*TP53INP2*	F: 5′-CCTCCCCTTCTCCTCCAGTAAA-3′ R: 5′-AGCCCAAAATTCAGTCTCACCA-3′
*FTO*	F: 5′-ACTTGGCTCCCTTATCTGACC-3′ R: 5′-TGTGCAGTGTGAGAAAGGCTT-3′
*NPM1-mA*	F: 5′-TGGAGGTGGTAGCAAGGTTC-3′ R: 5′- CTTCCTCCACTGCCAGACAGA-3′
*NPM1-wt*	F: 5′-ACGGTCAGTTTAGGGGCTG-3′ R: 5′-CTGTGGAACCTTGCTACCACC-3′
*β-actin*	F: 5′-TAGTTGCGTTACACCCTTTCTTG-3′ R: 5′-TGCTGTCACCTTCACCGTTC-3′

F stands for forward; R stands for reverse; *NPM1*-*mA* stands for *NPM1* mutation type A; *NPM1*-*wt* stands for *NPM1* wild-type.

## Data Availability

Contact the author if need the datasets generated during and/or analyzed during the current study.

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
