# Peer review of "Cytoplasmic Expression of TP53INP2 Modulated by Demethylase FTO and Mutant NPM1 Promotes Autophagy in Leukemia Cells"

_ijms, 2023, doi:10.3390/ijms24021624_

Round 1

Reviewer 1 Report

In the manuscript, Huang and colleagues have demonstrated how cytoplasmic TP53INP2 promotes autophagy in AML. The authors have shown that FTO-mediated modification upregulates TP53INP2, which interacts with NPM1 mutants and delocalizes to the cytoplasm. TP53INP2 promotes the interaction of autophagy mediators LC3 and ATG7 and subsequently facilitates cell survival.

The manuscript is well-written. The experiments were designed and performed flawlessly to address the research questions. One query to address:

-          TP53INP1 is a close associate of TP53INP2 and one would wonder about the status of TP53INP1 in this model. Please mention the expression profile of TP53INP1 in section 2.1.

Reviewer 2 Report

In this manuscript Huang et al. have investigated the role of TP53INP2 in autophagy processes in leukemia. The work is well-presented and discussed. The study is well designed and performed, and statistical analysis is appropriate.

Some minor changes: when referring to genes is preferable to use italic.

Introduction should be better presented and the various paragraphs should be better link each others.
